# Stabilization of Ultrafine Iron Tailings with Acrylic–Styrene Copolymer for Sustainable Geotechnical Applications

**DOI:** 10.3390/polym17192624

**Published:** 2025-09-28

**Authors:** Matheus Machado Lopes, José Wilson dos Santos Ferreira, Michéle Dal Toé Casagrande

**Affiliations:** Department of Civil and Environmental Engineering, University of Brasilia, Federal District, Brasilia 70910-900, Brazil; machadomlopes@gmail.com (M.M.L.); jose.wilson@unb.br (J.W.d.S.F.)

**Keywords:** new geotechnical materials, key parameters, tailings-polymer composite, microtomography

## Abstract

Considerable research in recent years has examined the reuse of tailings; however, the lack of particle cohesion limits their application as construction materials. Therefore, this study assessed the stabilization of ultrafine iron ore tailings using an acrylic–styrene copolymer. Geotechnical characterization and polymer dosage, hydromechanical and microstructural tests were carried out, including unconfined compressive strength (UCS), permeability, scanning electron microscopy (SEM) and microtomography (μCT). The polymer effectively enhanced the mechanical behavior of the tailings, increasing the UCS from 49 kPa for untreated material to 2114 kPa and 3324 kPa for 30% and 40% polymer content, respectively. A robust power-law model (R^2^ ≥ 0.90), based on the porosity/volumetric polymer index (η/Pᵢᵥ), was developed to predict strength, showing that mechanical gains can be achieved by increasing either polymer content or dry density, as supported by statistical analyses. Permeability remained on the order of 10^−6^ cm/s regardless of polymer addition, indicating that the polymer does not fill voids but instead acts as a binding agent, as confirmed by SEM and μCT analyses. Overall, this study establishes a technically feasible and sustainable approach for tailings management, highlighting the potential of polymer stabilization to turn environmentally challenging tailings into functional geotechnical materials.

## 1. Introduction

The mining industry accounts for 4.2% of Brazil’s Gross Domestic Product, GDP, according to the National Mining Plan—PNM. In order to be exported or even absorbed by domestic industry, minerals must undergo treatment or beneficiation processes, which consist of several unit operations and, finally, the disposal of tailings [1].

Regarding the final disposal, tailings have become an increasingly significant global challenge, as the most common methods—such as dams or piles—require vast land areas. This can conflict with land use and occupation issues, especially environmental concerns. Moreover, when these structures are built improperly, they pose serious risks to society, the environment, and the economy, as seen in the tailings dam disasters in Mariana and Brumadinho, in the state of Minas Gerais, in 2015 and 2019, respectively [2,3].

New methodologies are required for tailings disposal, by improving beneficiation efficiency to generate less waste, by developing safer technologies for disposal, or by providing alternative uses for these materials.

The reuse of tailings as construction material has been widely investigated in recent years; however, their mechanical properties are often inadequate for engineering applications, making property enhancement necessary. One efficient way to improve the properties of tailings is through the incorporation of stabilizing agents. The most commonly used agents applied in stabilization techniques include cement, lime, fly ash, asphalt emulsion, and construction and demolition residues, according to Ferreira et al. [4].

In this scenario, polymer stabilization has emerged as a promising alternative. Polymers act as physical binding agents, coating particles and forming a flexible, three-dimensional network. Recent studies have demonstrated their efficacy, such as Carneiro et al. [5], Alelvan et al. [6] and Boaventura et al. [7]. All three studies demonstrated the feasibility of using the polymers as a chemical stabilizer, with significant improvements in the material’s mechanical strength. Carneiro et al. [5] showed significant shear strength gains in iron tailings and discarded the theory that the resistance gain of the composite was only due to a suction increase for longer curing days, since suction was kept nearly constant and the values of suction were close to those of the tailings without additive. Alelvan et al. [6] successfully stabilized gold tailings, noting improved ductility and an increase in the unconfined compression strength. Through SEM analysis, they further confirmed that the polymer interacts with the lamellar tailing particles to form aggregates, a process that can be regarded as a type of artificial cohesion, which leads to a marked reduction in the porosity of the matrix. Boaventura et al. [7] further demonstrated the viability of synthetic polymers for sandy soils, as the polymer enhanced cementation and cohesion to the substrate. From an environmental perspective, they also reported that the chemical analysis did not indicate excessive levels of elements that could contaminate groundwater, fauna, flora or the local population, thereby underscoring the potential environmental benefits of this approach. The key advantages of polymers include improved ductility, increased strength and the potential to achieve a lower carbon footprint compared with conventional cementitious binders.

As highlighted in the literature, compressive strength and indirect tensile strength are the primary parameters used to evaluate the mechanical behavior of stabilized geotechnical materials. In this context, a conceptual framework for understanding strength development for cement-stabilized soils was proposed by Consoli et al. [8]. Two key factors play a fundamental role: the void ratio, expressed through porosity—η (Equation (1)); the volumetric cement content—*C*_iv_ (Equation (2)).(1)$$\eta =100-100\cdot \frac{\left\{\left[\frac{C\cdot \rho_{d}\cdot V_{T}}{\left(1\;+\;C\right)}\cdot \frac{1}{d_{c}}\right]+\left[\frac{\gamma_{d}\cdot V_{T}}{\left(1\;+\;C\right)}\cdot \frac{1}{\rho_{SR}}\right]\right\}}{V_{T}}$$
where C is the cement amount in the mass of dry soil; $\rho_{d}$ is the specific mas of the sample; $d_{c}$ is the density of the cement agent; $\rho_{SR}$ is the dry unit mass of soil grains; and $V_{T}$ is the total volume of the sample.

To evaluate the influence of the cementing agent content on a single index, a new index called volumetric cement content ($C_{iv}$), was proposed, as follows:(2)Civ = VCVT = mCdCVT
where $V_{C}$ is the cement agent volume; $V_{T}$ is the total volume of the sample; $m_{C}$ is the cement agent mass; and $d_{C}$ is the density of the cement agent. The general equation for both the compressive and tensile strengths relate *η*/$C_{iv}$ to the curing time ($A_{i}$), along with the adjustment exponents β and B (Equation (3)).(3)$$q_{u}\;or\;q_{t}={\left[\frac{\eta }{{{(C}_{iv})}^{\beta }}\right]}^{-B}$$

Consoli et al. [9] investigated the adequacy of the porosity/cement index (η/$C_{iv}$) in predicting the elastic and plastic characteristics of compacted filtered iron ore tailings–Portland cement blends, proving its efficiency for field control and determining the dosage of iron ore tailings–Portland cement mixtures for application in dry stacks.

Despite these promising results, substantial knowledge gaps remain concerning the effectiveness of applying this framework to tailings stabilization with polymers, as the interactions between tailings and polymeric stabilizers are complex and not yet fully understood. The optimal dosage for different tailings remains unpredictable, and the microstructural mechanisms governing strength and deformation behavior are often inferred rather than quantitatively measured. Furthermore, although the porosity/binder index framework is a powerful tool for cemented soils, its applicability to polymer-stabilized tailings has not yet been rigorously validated.

Particularly, microstructural analyses using scanning electron microscopy (SEM) and X-ray microtomography (μCT) provide valuable insights [10]. These techniques enable researchers to examine the internal behavior of geomaterials in a non-invasive and non-destructive manner. Moreover, they require only small sample sizes while providing critical information for validating theories and advancing understanding of tailings mechanics [6,11,12].

Thus, this study aims to investigate the stabilization of ultrafine iron tailings using an acrylic–styrene copolymer. This polymer was selected for its anionic character; high solubility in water, which ensures homogeneous mixing; and ability to form a flexible, cohesive film upon curing. The objectives are to (1) determine the optimal polymer dosage for maximizing strength and ductility; (2) assess the synergistic effect of dry density and polymer content; (3) develop a predictive model for the strength of polymer-stabilized tailings based on the porosity/volumetric binder index (η/P_iv_) concept; and (4) reveal the microstructural mechanisms driving enhanced macroscopic behavior. By integrating mechanical and microstructural analyses within a predictive framework, this work establishes polymer stabilization as a practical, sustainable and highly effective strategy for transforming tailings into viable geotechnical materials.

## 2. Materials and Methods

Figure 1 presents a flowchart of the material selection process and the sequential methods applied in this research. Based on the tailings characterization, preliminary compaction and mix design tests were performed to establish the optimal conditions for the subsequent hydromechanical and microstructural analyses.

### 2.1. Materials

The ultrafine iron tailings (Figure 2a) originate from the Maravilhas II dam mine, located at Mina do Pico in the city of Itabirito, Minas Gerais. This mine is one of the main dams in the complex; it is of the downstream type, implemented in 1994, holds a volume of 94 million cubic meters, is 97.9 m high, and currently has 109 monitoring instruments [13].

The polymer stabilizer used in this work (Figure 2b) was provided by Waterflows Biochemicals, Mogi Mirim, Brazil. It consists of an organic acrylic–styrene copolymer and is presented in the form of an aqueous emulsion of anionic character. It has a pH of 8.0–9.0, density of 0.98–1.04 g/cm^3^, and viscosity of 3000–10,000 centipoise (cP). In addition, it is completely soluble in water.

### 2.2. Characterization of the Tailings

The determination of the specific grain mass was carried out using the pentapycnometer equipment Quantachrome model Pentapyc 5200e from the Geotechnical Laboratory of the University of Brasília, Brasília, Brazil as established by [14]. The granulometric distribution curve was constructed following [15,16], with and without deflocculant, since there is no specific standard for tailings. Despite this, it is considered a good approach for carrying out this type of test to classify the particle size of tailings particles.

The X-ray diffraction (XRD) tests were carried out to identify the crystalline structure present in pure tailings from a sample passing through a 75 μm sieve and dried in an oven at 105 °C. The analysis was carried out on a Rigaku Ultima IV diffractometer from the X-ray Diffractometry Laboratory of the University of Brasília, Brasília, Brazil, under a voltage of 35 kV and 15 mA, an angular amplitude of 2θ, a measurement range between 2 and 100°, and a speed of 0.05°/min. Mineralogical identification was carried out using the reference standards of the Match! 4 software databases. Tests for the tailings–polymer composite were not performed since the polymer is organic and does not present a crystalline structure.

### 2.3. Tests of Tailings–Polymer Composite

Initially, an attempt was made to obtain the compaction curve of the tailings. However, it was observed that the addition of water produced a slurry that impaired and hindered the workability of the pure tailings. Consequently, the moistened material seeped through the gaps between the spacer disc and the mold, preventing the application of the standard procedure prescribed by the relevant standard. This finding underscores the challenge of applying the conventional soil mechanics methodological framework to tailings mechanics. Therefore, a moisture content of 15% and an initial dry density of 1.91 g/cm^3^ were set. Polymer dosages of 10%, 20%, 30%, 40%, 50%, and 60% were then tested to determine the optimal content for ultrafine iron tailings from the perspective of mechanical behavior.

The polymer percentages were calculated from the percentage of water added to the tailings to dilute the polymer and facilitate homogeneity. Accordingly, the effective polymer content in the composite was determined proportionally to the 15% moisture content. The experimental conditions investigated, the effective polymer percentages, and the abbreviations used in this study are summarized in Table 1.

Samples measuring 100 mm in height and 50 mm in diameter were molded, in triplicate, for each experimental condition. The composites were air-cured for 14 days before the unconfined compressive strength (UCS) tests, based on previous work [5,6,7]. The curing period for dosage was established taking into account that resistance gains are typically observed after 14 days.

### 2.4. Unconfined Compressive Strength Tests

From the dosage tests, the concentrations of 30% and 40% were identified as more effective for the mechanical tests. To expand the understanding of the behavior of tailings–polymer composites, keeping the content fixed, the composites’ dry densities were varied across 1.91, 1.96, and 2.01 g/cm^3^. Then, samples in triplicate with 100 mm height and 50 mm diameter for pure tailings and the tailings–polymer composite were molded and air-cured for 28 days. The curing period was established considering that most reactions occur within this period; longer periods present minimal impact on strength gains, as seen from [5]. After that, the mechanical behavior was determined through UCS tests [17].

The tests occurred under a controlled deformation of 1.27 mm/min up to 10% of deformation. As an acceptance criterion for the UCS tests, it was established that the strength of the replicates, molded with the same characteristics, should not deviate more than 10% from the average strength.

To evaluate the statistical significance of variations in unconfined compressive strength as a function of the studied factors—polymer concentration and dry density—an analysis of variance (ANOVA) was conducted. This procedure tests the null hypothesis that all group means are equal against the alternative that at least one differs. Statistical significance was established at a 5% significance level (α = 0.05). Additionally, Tukey’s post hoc test was performed to identify which specific groups exhibited significant differences.

### 2.5. Permeability Tests

The evaluation of hydraulic behavior is essential in chemical stabilization studies with tailings for geotechnical purposes, since there may be the presence of heavy metals in the material. Additionally to mechanical behavior, it is fundamental that the proposed solution retains or minimizes the potential for leaching if any of these elements are present.

Thus, permeability tests were carried out for pure tailings and 40% polymer-stabilized tailings for a dry density of 1.91, 1.96, and 2.01 g/cm^3^ and a curing time of 28 days to evaluate the influence of the polymer and the density on the permeability coefficient. The permeability tests were conducted using the falling-head method based on [18]. Water flowed through the specimen from a standpipe. Following specimen saturation over two days, the reservoir was filled, and water discharge was initiated. The water column height was measured at intervals of over 24 h, with a total of 14 readings per specimen, to obtain a representative average coefficient of permeability (k) for each experimental condition.

### 2.6. Microstructural Tests

The microstructure of the pure tailings and tailings–polymer composites was assessed by scanning electron microscopy (SEM) and X-ray microtomography (μCT) tests. SEM was performed using a JEOL JSM-7001F microscope from the Microscopy and Microanalysis Laboratory of the University of Brasília, Brasília, Brazil, which has a field emission scanning electron microscope (FE-SEM) with a hot electron gun (Schottky) optimized for analytical applications. The tests were operated under an acceleration voltage of 5 kV. Both the pure tailings and composite were evaluated using cylindrical specimens with 30 mm height and 15 mm diameter. The ultrafine iron tailings sample was prepared by passing tailings through a 75 μm sieve and dried in an oven at 105 °C for 24 h before analysis. For the analysis of the tailings–polymer composite, a compacted cylinder 30 mm high and 15 mm in diameter was prepared.

X-ray microtomography (μCT) analysis is a powerful microstructural technique for determining the effect of the polymer in the tailings matrix and the three-dimensional model of the composite. The experimental conditions evaluated in the test were pure tailings and tailings stabilized with 40% polymer, both at a 1.91 g/cm^3^ dry density, with a 30 mm height and 15 mm diameter (Figure 3). From the two-dimensional image (Figure 3a), it can be observed that the regions closest to the mold exhibit lower density, represented by the colors blue, green, and yellow. Toward the center, higher density and greater homogeneity are indicated by the purple color. To minimize edge effects, data acquisition was therefore carried out within a volume of interest (VOI) measuring 12.5 mm in diameter and 12.5 cm in height (Figure 3b).

The cylindrical samples were analyzed using a Bruker SkyScan 1172 MicroCT system from the Laboratory of Physical Properties of Rocks at the University of Brasília, Brasília, Brazil throughout 666 images. They were obtained with a 3150 ms exposure time using an aluminum filter of 0.5 mm. They had an imaging resolution of 4.96 μm, with a voltage and electric current of 70 kV and 129 μA for all the investigated materials. The reconstruction of the three-dimensional model was performed in the NRecon V 1.7 software.

## 3. Results and Discussion

### 3.1. Ultrafine Iron Tailings Characterization

Figure 4a presents the mineral composition of ultrafine iron ore tailings.

The results show that the iron tailings studied are predominantly composed of quartz [SiO_2_] and hematite [Fe_2_O_3_]. These results are similar to others obtained in studies carried out on distinct iron ore tailings such as [5,19]. The tailings specific gravity is 3.23 g/cm^3^; this value is slightly higher than that from the literature since the present material is ultrafine, which leads to a greater quantity of physically heavy mineral particles, such as hematite, by volume when compared to the other work [5,19,20].

Figure 4b presents the particle size distribution of the material. It can be observed that the iron tailings predominantly consist of the sand-size fraction (92.7%). However, despite this classification, the material does not exhibit the typical behavior of a soil.

### 3.2. Dosage Tests of Tailings–Polymer Composite

Figure 5 shows the unconfined compressive strength results for different polymer dosages after 14 days of air-curing. From the stress–strain curves, it can be noted that the addition of polymer improves the compressive strength of the tailings. The gains increase with the addition of polymer by up to 40%, after which the compressive strength peaks tend to present smaller magnitudes. It is important to note that compacted pure tailings exhibited a stress peak of 49 kPa under 0.05% axial strain, which was not plotted due to the scale magnitude.

The results demonstrated a satisfactory physical–chemical interaction of the ultrafine iron tailings with the polymer, which is shown by the greater load-bearing capacity of the composites. In addition to a significant gain in the material’s peak strength, the results showed satisfactory ductility when compared to other tailings–polymer composites, considering that the peak occurred at axial strain from 4% to 8%, while the composites studied by [5,6] presented their peaks at 1–4% axial strain. This difference may be related to the interaction between the particle size and the polymer, indicating that materials with smaller particle sizes may present better performance. After the peak, an abrupt drop in strength is observed, indicating that the polymer stabilizing the tailings is not effective in post-peak resistance.

Considering that the polymer acts as a glue effect, joining the particles, once the peak is reached, it is expected that this cementing effect will break [6]. If post-peak resistance is an important factor in design, studies such as Sotomayor et al. [20,21], Festugato et al. [22] and Consoli et al. [23] are relevant because they inserted fibers in stabilized tailings to ensure a better distribution of stresses and reduce the drastic post-peak drop.

Table 2 shows the peak stress values for each composite as well as the percentage of resistance gain compared to the compacted pure tailings. From the results, it is evident that the optimum dosage content was a 40% concentration of the polymer solution. Nevertheless, composites with 30% polymer also exhibited satisfactory results and represent a less costly solution, thereby justifying the study of both concentrations. It is important to point out that particle shape and size, and mineralogical composition affect the optimum polymeric content, as indicated by Alelvan et al. [6] and Boaventura et al. [7].

### 3.3. UCS

From the stress–strain curves of tailings stabilized with 30% (Figure 6a) and 40% polymer (Figure 6b), it is seen that both polymer content and dry density affected the mechanical behavior of the material. It should be emphasized that the curves presented correspond to the mean responses of three specimens for each experimental condition, thereby facilitating the visualization of the overall trend.

The increase in density from 1.91 to 1.96 and 2.01 g/cm^3^ positively influenced the peak resistances presented by both contents, improving that in T-P30 from 3633.3 kPa to 3890.7 kPa and 4124.0 kPa, respectively. Considering the T-P40 composites, the values increased from 4242.3 kPa (T-P40/1.91) to 4689.7 kPa (T-P40/1.96) and 5406.0 kPa (T-P40/2.01). Still, comparing the effects of increasing the polymer content (Table 2) with increasing the dry density, the first has a more pronounced effect on the mechanical behavior of the composite.

Table 3 summarizes the means ± standard errors of the UCS values for all the experimental conditions. The analysis of variance (ANOVA) indicated that the differences between treatments were statistically significant (*p*-value < 0.05). Based on this finding, Tukey’s post hoc test was performed to identify which treatments differed significantly, with the results indicated by distinct superscript letters in Table 3.

The statistical analysis indicates that increasing the dry unit weight to 2.01 g/cm^3^ for the 30% polymer composite (T-P30/2.01) yields an unconfined compressive strength comparable to that of the 40% polymer composite at 1.91 g/cm^3^. This equivalence suggests that, from a practical perspective, an evaluation of which parameter—polymer content or dry unit weight—minimizes project cost is warranted. Moreover, increasing the dry unit weight from 1.96 to 2.01 g/cm^3^ did not produce statistically significant differences, suggesting a mechanical response threshold at a 30% polymer dosage. In contrast, for the 40% polymer composites, the mean UCS values differed significantly across the three dry unit weight treatments, indicating well-defined performance tiers under the tested conditions.

To highlight the effect of the polymer on the mechanical behavior of tailings, the tenacity of the composites was assessed according to the strain energy absorption capacity (E_def_), which is a quantity numerically equal to the area under the stress vs. axial strain curve, evaluated up to the peak strength (Figure 7). It is shown that, the greater the amount of polymer in the composite, the greater the material’s ability to absorb energy up to peak strength, which is in agreement with Alelvan et al. [6]. Particularly regarding dry densities, it is observed that the increase from 1.91 to 1.96 yields considerable gains, i.e., 20.6% and 69.9% for T-P30 and T-P40, respectively. However, the increase from 1.96 to 2.01 does not result in significant magnitudes of improvement.

Based on Equations (1) and (2), the unconfined compressive strengths of tailings–polymer composites as a function of the porosity/volumetric polymer content index (η/P_iv_) are demonstrated in Figure 8. Overall, the results demonstrate that lower porosity of the composite corresponds to higher UCS. In particular, it is important to highlight that grouping the results by the dry mass of the composites leads to an overlap of the trend curves, showing that identical resistance values can be achieved either by increasing the polymeric solution or by increasing the density. This finding corroborates the previously presented statistical results.

From mathematical adjustments (Figure 8a), an external exponent equal to 0.28 was applied in P_iv_ to fit the data accordingly. The external exponent value used was identical to that of Consoli et al. [8,24] and Scheuermann Filho and Consoli [25], who used lime and cement fly ash to stabilize residual soil. As the external exponent expresses the relative importance of the porosity and the amount of polymeric solution, as noted by [26], the adjustment indicates that porosity remains as the main factor in the strength of the mixtures regardless of whether the matrix is soil or tailings.

By fitting results in a power function, satisfactory correlations were obtained (R^2^ ≥ 0.90), in which A and B are scalars related to the matrix characteristics and nature of the stabilizing agent, and the first is also strongly associated with the curing time.

Lastly, a method for the data normalization of compressive strength is proposed in Figure 8b by dividing both the left and right sides of the equations to make the mechanical behavior from q_u_ of η/P_iv_ equal to 60 (Equation (3)), an intermediate value within the range from 50 to 70 in this study. Additional details of the mathematical normalization process can be found in the relevant literature [6,25].

With this analytical approach, is possible to determine the required resistance from a single equation using just one experimental test with a molded sample of a specific polymer quantity and density, at a particular porosity. This saves time and materials, and makes the analysis more efficient. It is important to clarify that the investigation of mechanical behavior is restricted to the scope of the experimental program.

### 3.4. Permeability

The permeability coefficient obtained for the pure tailings and tailings composites with 40% polymer is shown in Table 4. Overall, no significant change in *k* was observed between the pure and stabilized tailings, keeping the order of magnitude practically constant across all the experimental conditions investigated ($10^{-6}$ cm/s). For identical dry densities, the addition of polymer slightly reduces the permeability coefficient due to the agglomeration effect among the ultrafine tailings’ particles. Nevertheless, the amount of polymer used is insufficient to induce significant changes in the permeability property, unlike the observed effects on mechanical behavior.

### 3.5. Microstructure

Figure 9 shows the SEM images of pure tailings and the composites T-P40-1.91, T-P40-1.96, and T-P40-2.01 magnified at 500×. Ultrafine iron tailings particles present varied particle sizes with an emphasis on their angular shape, corroborating the results of the granulometric analysis. On the other hand, the addition of polymer improves the homogeneity in the distribution of particles along the investigated surface.

Considering that both pure tailings and T-P40-1.91 exhibit the same density and, consequently, porosity, the SEM images (Figure 9a,b) reveal a difference in the size of the voids. Unlike cement, where reactions fill the spaces [27], it is hypothesized that polymer stabilization alone does not fill these voids. Instead, its effect lies in the agglutination of the particles, which reduces the size of the macropores and increases their relative frequency. Consequently, there is an increase in the contact points between the particles, leading to improved load distribution when stressed, as evidenced by the obtained mechanical properties.

With respect to the increase in density in the tailings composites stabilized with 40% polymer, it is evident that higher densities promote closer particle packing, thereby enhancing the polymer’s binding effect. This phenomenon is evident in the T-P40-2.01 composite (Figure 9d), where a substantial area exhibiting this effect can be delimited.

The three-dimensional μCT analysis of the pure tailings and the tailings–polymer composite with 40% polymer (T-P40) is summarized in Table 5, which reports the porosity, tailings grains, tailings–polymer aggregates, and denser minerals (shown in blue). 

The presence of the polymer introduces an additional category, likely corresponding to the aggregation of tailings grains with the polymer, as indicated by the higher density detected in the histogram. Table 5 also presents the relative percentage of each category derived from the total volume of the specimen. Notably, the porosity of the 40% polymer composite decreased by 1.78% compared with that of the pure tailings, reinforcing the evidence that the polymer contributes to pore closure within the composite structure.

## 4. Conclusions

The acrylic–styrene copolymer proved highly effective in enhancing the mechanical properties of the tailings. An optimal polymer dosage range of 30–40% was identified, yielding an increase in strength from 49 kPa for untreated tailings to 3324 kPa for tailings stabilized with 40% polymer. Beyond this optimum, the strength gains diminished, indicating an economic and technical threshold for stabilization. In addition, the polymer significantly improved the material’s ductility and energy absorption capacity, transforming the brittle tailings into a more compliant composite—an especially desirable property in geotechnical applications to prevent failures at low levels of deformation.

A major contribution of this work is the successful adaptation of the porosity/volumetric polymer index (η/Pᵢᵥ) to model the strength of polymer-stabilized tailings. The development of a robust power-law model (R^2^ ≥ 0.90) provides a valuable predictive tool for engineers. This model demonstrates that strength can be enhanced by increasing the polymer content or by increasing dry density, offering flexibility in design and optimization, which was confirmed by statistical analysis. Such an approach substantially reduces the need for extensive laboratory testing, enabling the efficient estimation of composite strength based on a limited set of parameters.

Microstructural analyses from SEM and X-ray microtomography provided critical insights into the stabilization mechanism. The polymer acts primarily as a binding agent, promoting particle agglomeration and reducing macroporosity by 1.78%. This process enhances matrix homogeneity and increases inter-particle contact points, resulting in more effective load transfer and stress distribution. It is concluded that the strength improvement arises not from pore-filling but from targeted modification of the pore structure and inter-particle bonding.

Importantly, the polymer stabilization process did not adversely affect the hydraulic conductivity of the tailings, with permeability remaining on the order of 10^−6^ cm/s. This indicates that the proposed method does not increase the leaching potential of contaminants, addressing a key environmental concern in tailings reuse. Overall, this research establishes a technically feasible and sustainable pathway for the valorization of ultrafine iron ore tailings, transforming an environmentally challenging waste product into a viable engineered material for civil construction applications, such as backfill, road base, or sub-base layers.

## Figures and Tables

**Figure 1 polymers-17-02624-f001:**
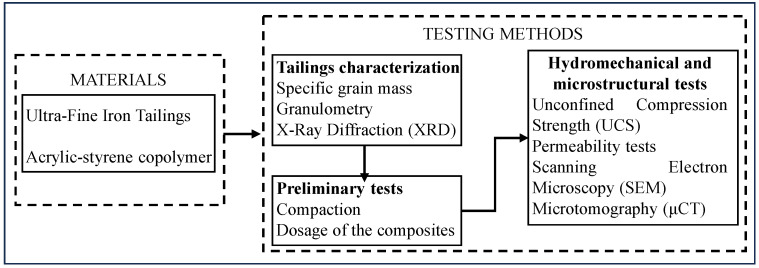
Flowchart of the experimental program.

**Figure 2 polymers-17-02624-f002:**
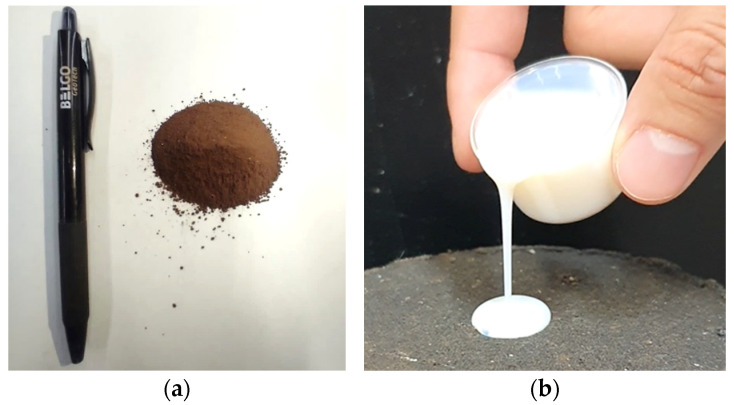
Materials used: (**a**) ultrafine iron tailings; (**b**) polymer.

**Figure 3 polymers-17-02624-f003:**
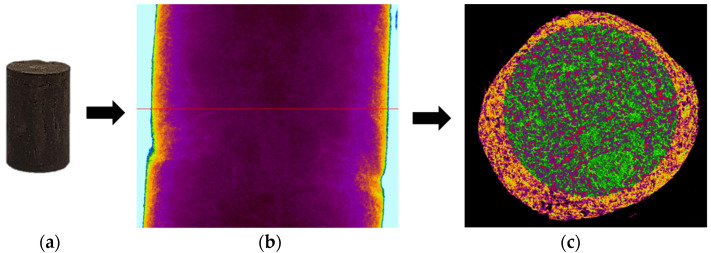
Region of interest in μCT: (**a**) sample; (**b**) two-dimensional image; (**c**) section area.

**Figure 4 polymers-17-02624-f004:**
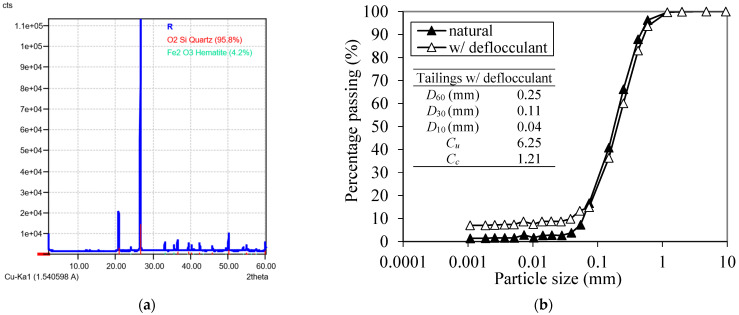
Pure tailings: (**a**) XRD; (**b**) particle-size distribution curve.

**Figure 5 polymers-17-02624-f005:**
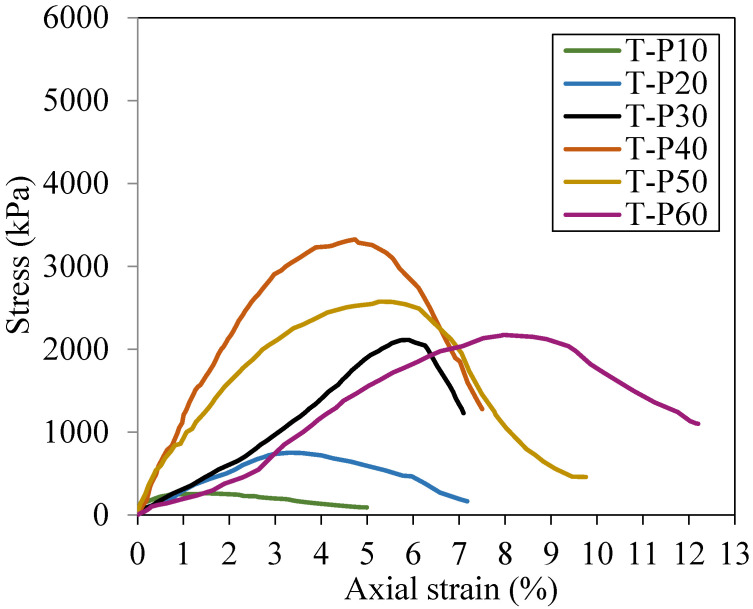
Stress–strain curves for different dosages after 14 days of curing.

**Figure 6 polymers-17-02624-f006:**
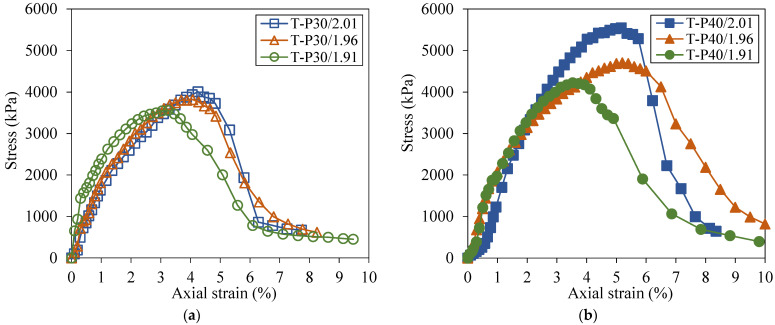
Tailings–polymer composites under distinct dry unit masses, with polymer dosages of (**a**) 30% and (**b**) 40%.

**Figure 7 polymers-17-02624-f007:**
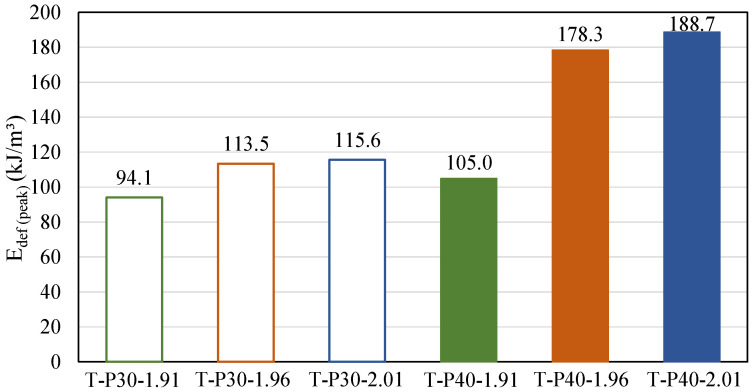
Strain energy absorption capacity of tailings–polymer composites.

**Figure 8 polymers-17-02624-f008:**
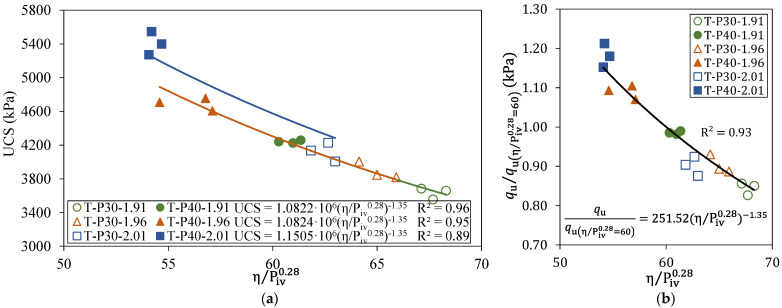
Effect of porosity/polymer ratio (η/P_iv_) on the strength of tailings–polymer composites: (**a**) fitting equations; (**b**) normalization of *q*_u_ at η/P_iv_ = 60.

**Figure 9 polymers-17-02624-f009:**
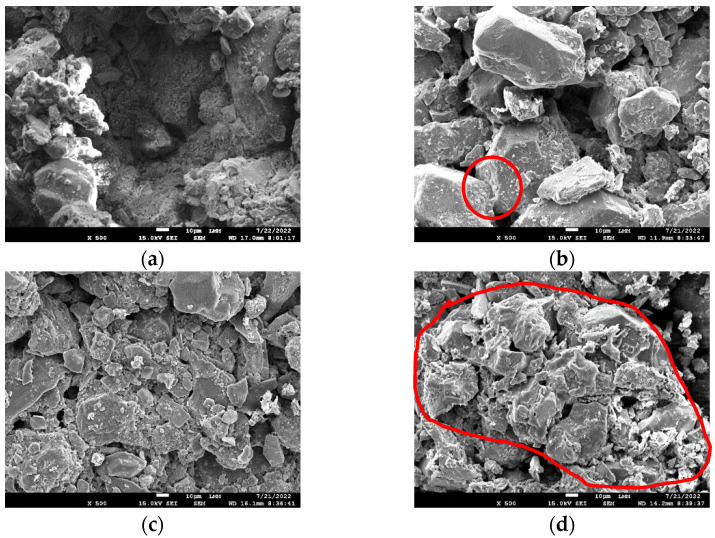
SEM images magnified at 500×: (**a**) pure tailings; (**b**) T-P40/1.91; (**c**) T-P40/1.96; (**d**) T-P40/2.01.

**Table 1 polymers-17-02624-t001:** Experimental conditions of tailings–polymer dosage.

Relative Percentage (%)	Effective Percentage (%)	Abbreviation
10	1.5	T-P10
20	3.0	T-P20
30	4.5	T-P30
40	6.0	T-P40
50	7.5	T-P50
60	9.0	T-P60

**Table 2 polymers-17-02624-t002:** Strength development of the composites as a function of polymer content.

Composite	Peak Strength (kPa)	Strength Gain (%)
Pure Tailings	49	---
T-P10	263	537
T-P20	753	1537
T-P30	2114	4314
T-P40	3324	6784
T-P50	2574	5253
T-P60	2175	4439

**Table 3 polymers-17-02624-t003:** Mean unconfined compressive strength and statistical analysis.

Treatment	UCS_mean_ (kPa)
T-P30/1.91	3633.3 ± 38.8 **^e^**
T-P30/1.96	3890.7 ± 58.3 **^d^**
T-P30/2.01	4124.0 ± 64.0 **^cd^**
T-P40/1.91	4242.3 ± 9.83 **^c^**
T-P40/1.96	4689.7 ± 44.2 **^b^**
T-P40/2.01	5406.0 ± 79.5 **^a^**

**Table 4 polymers-17-02624-t004:** Permeability coefficient (k) of pure and polymer-stabilized tailings.

Experimental Condition	k (cm/s)
T-1.91	1.2 × 10^−5^
T-1.96	9.5 × 10^−6^
T-2.01	6.6 × 10^−6^
T-P40-1.91	8.4 × 10^−6^
T-P40-1.96	5.5 × 10^−6^
T-P40-2.01	2.7 × 10^−6^

**Table 5 polymers-17-02624-t005:** Results of X-ray micrography (μCT) in relation to sample volume.

Condition	Porosity (%)	Tailings (%)	Tailings–Polymer Aggregates (%)	Denser Minerals (%)
	21.13	78.53		0.32
Pure tailings	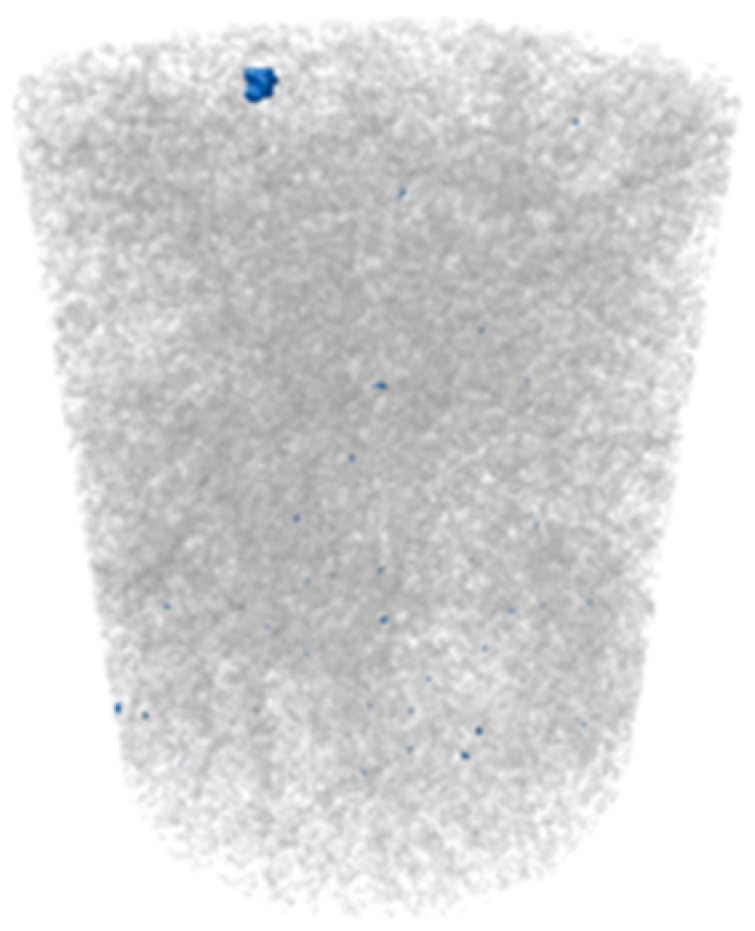	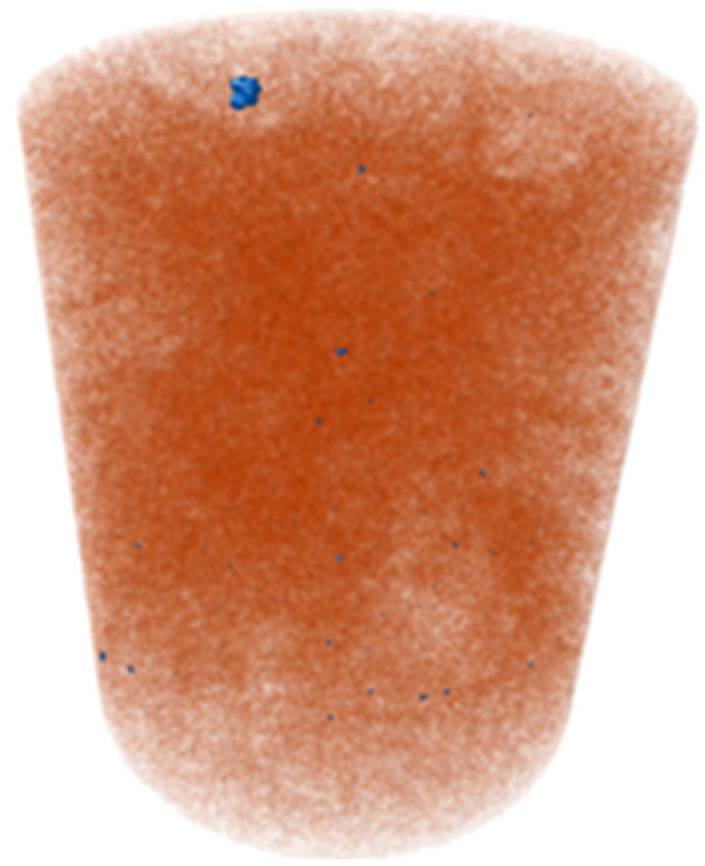	-	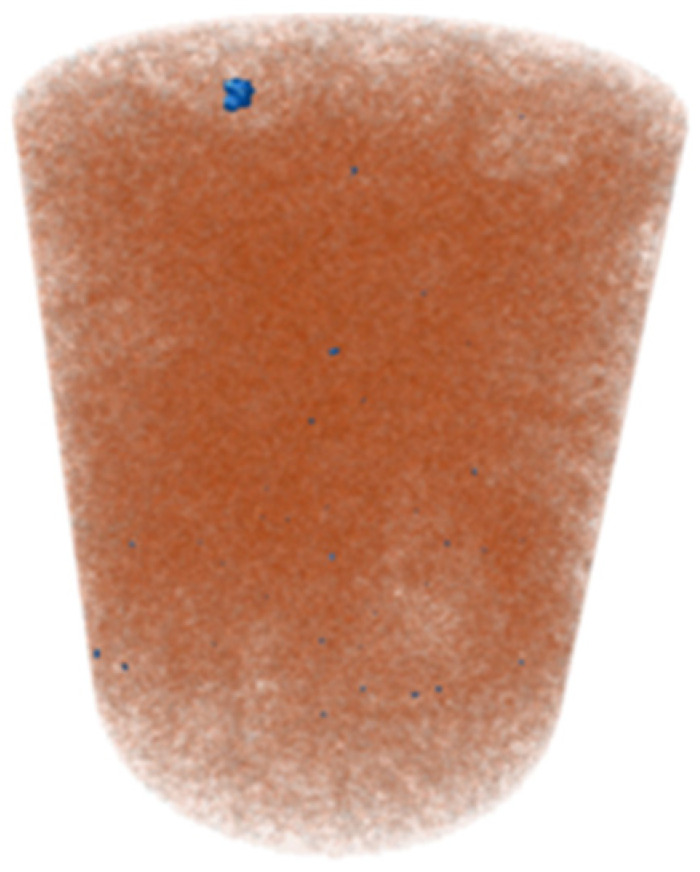
T-P40	19.35 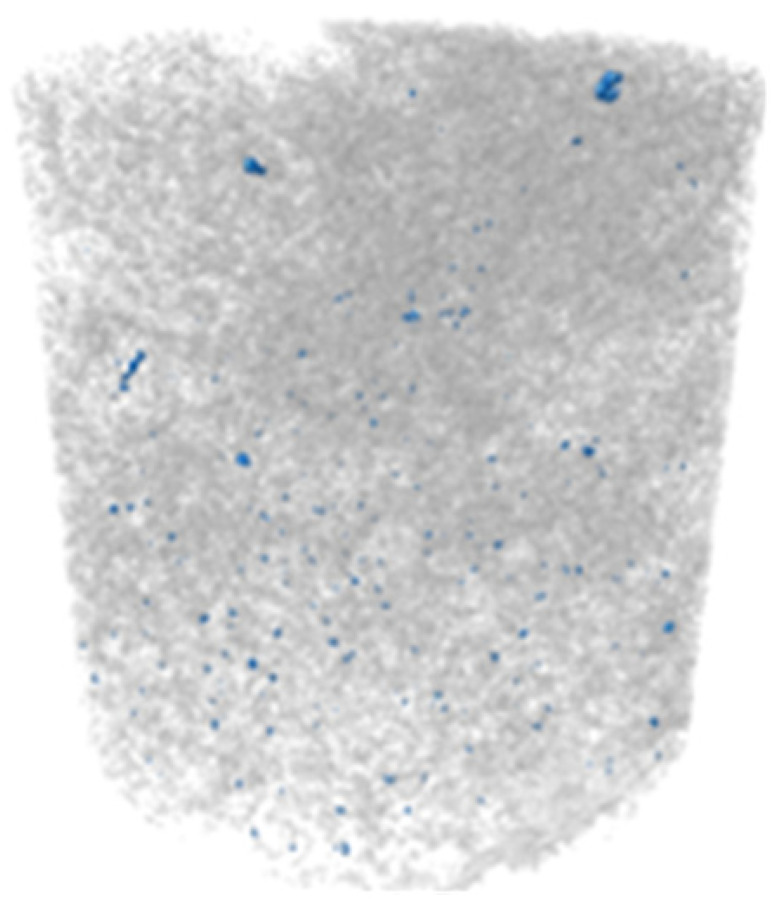	47.84 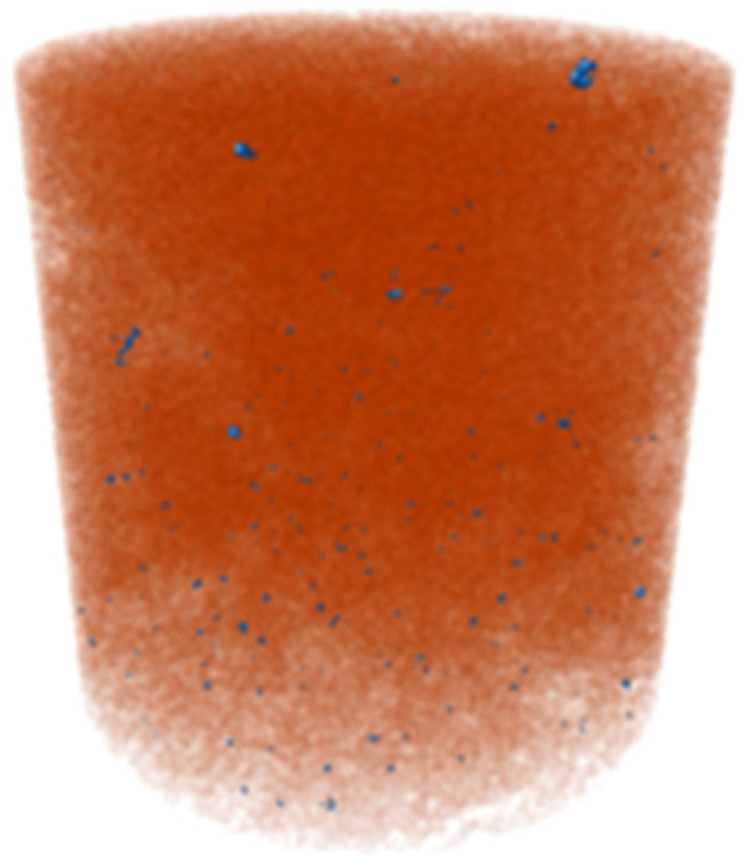	31.03 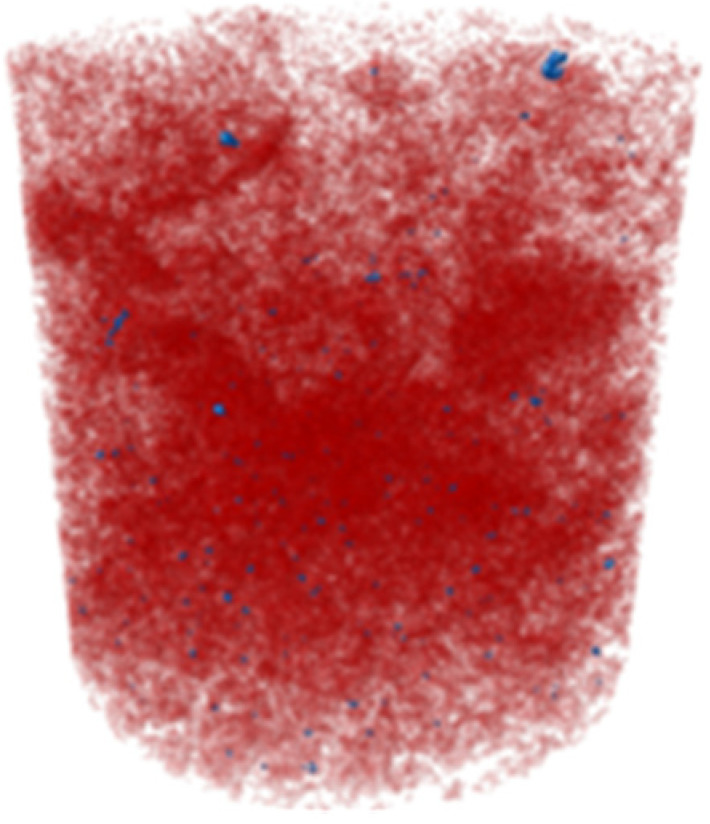	0.79 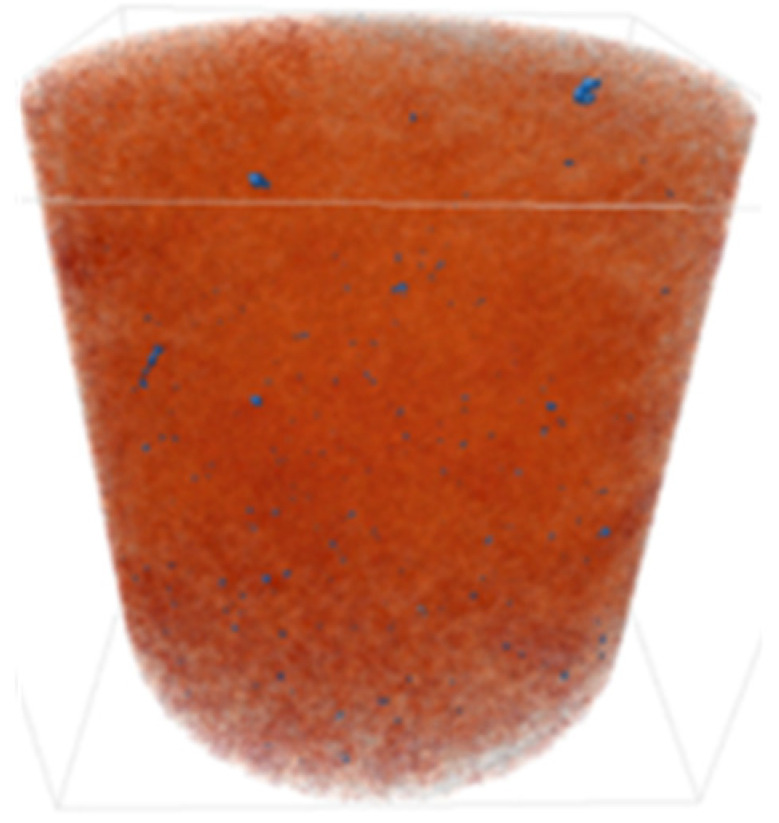

## Data Availability

The original contributions presented in this study are included in the article. Further inquiries can be directed to the corresponding author.

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
