# Peer review of "Stabilization of Ultrafine Iron Tailings with Acrylic–Styrene Copolymer for Sustainable Geotechnical Applications"

_polymers, 2025, doi:10.3390/polym17192624_

Round 1

Reviewer 1 Report

Comments and Suggestions for Authors

I have reviewed the paper “Challenges in the stabilization of ultrafine iron ore tailings using polymeric solution” and I recommend this paper for publication after major revision.

  • The title of the paper should be revised since the present one looks like review article title.
  • The abstract section may be shortened
  • The literature review should be strengthened and the novelty should be highlighted.
  • Check all the figure captions. The figure captions should be in detail
  • Figure 3. Region of Interest in μCT – scale mark is missing
  • Table 1. Experimental conditions of tailing-polymer dosage – how the percentage is fixed. Justification is required
  • Ensure for all the tests, the standard is mentioned
  • For microstructural tests- the scale mark is essential
  • 5a presents the mineral composition of ultrafine iron ore tailings – figure caption for 5a is missing
  • 5a presents the mineral composition of ultrafine iron ore tailings- the redraw the pattern using suitable software and present
  • Figure 7. Stress-strain curves for different dosages at 14 curing days – no need of markers on the curves
  • The reason for the improvement in strength properties should be explained in detail with literature support

Author Response

Comment

Response

The title of the paper should be revised since the present one looks like review article title.

The authors agree with the suggestions/corrections and these have been incorporated into the text of the paper.

The suggestion was accepted and the paper title was changed from “Challenges in the stabilization of ultrafine iron ore tailings using polymeric solution” to “Stabilization of Ultrafine Iron Tailings with Acrylic-Styrene Copolymer for Sustainable Geotechnical Applications”, which the authors believe synthesizes the proposed work more effectively.

The abstract section may be shortened

The authors agree with the suggestions/corrections and these have been incorporated into the text of the paper.

We thank the reviewer for this suggestion to improve conciseness. The abstract has been significantly shortened to less than 200 words, according to the journal’s standard.

The literature review should be strengthened and the novelty should be highlighted.

The authors agree with the suggestions/corrections and these have been incorporated into the text of the paper.

The literature review has been revised and strengthened to highlight the innovations proposed in this study, including the challenge of using polymers to stabilize ultrafine tailings, the development of a predictive strength model based on the porosity/volumetric binder index (η/Pᵢᵥ) concept—which until now has been applied primarily to cement-stabilized soils—and the demonstration of microstructural stabilization mechanisms between the polymer and tailings using SEM and μCT. Such microstructural investigations are scarce in the literature, as most studies focus solely on macroscopic aspects.

Check all the figure captions. The figure captions should be in detail

The authors agree with the suggestions/corrections and these have been incorporated into the text of the paper.

All figure captions were reviewed, and major corrections were applied where necessary.

Figure 3. Region of Interest in μCT – scale mark is missing

The images indicating the regions of interest in the μCT scans were automatically generated by the processing software, without a scale bar and only with a color-based visual indicator. Therefore, the authors chose to present the images faithfully as generated. Nevertheless, the information regarding the volume of interest (VOI), measuring 12.5 mm in diameter and 12.5 mm in height, is provided in the text discussing Figure 3, which the authors believe is sufficient for the interpretation of the images.

Table 1. Experimental conditions of tailing-polymer dosage – how the percentage is fixed. Justification is required

The experimental conditions for the dosages of polymer-stabilized tailings were established based on previous studies involving polymer-stabilized soils (Barreto et al., 2022; Boaventura et al., 2023), as well as prior research on the stabilization of iron ore and gold tailings with polymers (Alelvan et al., 2022; Carneiro et al., 2025). Drawing on this theoretical foundation and preliminary tests, a working range of 10–60% polymer, relative to the amount of water to be added, was defined.

Ensure for all the tests, the standard is mentioned

The authors agree with the suggestions/corrections and these have been incorporated into the text of the paper.

All standards on which the tests were based are cited in the text. In the specific cases of XRD, SEM, and μCT, the experimental procedures adopted for their execution are described in detail.

For microstructural tests- the scale mark is essential

In the case of the microstructural results from scanning electron microscopy (SEM), a scale bar is presented, automatically generated during image processing. For μCT, however, the post-processing and image generation did not provide a scale bar. Nevertheless, as mentioned in the methodology section, the tests were conducted within a volume of interest (VOI) measuring 12.5 mm in diameter and 12.5 mm in height, thereby providing a reference for image scale.

5a presents the mineral composition of ultrafine iron ore tailings – figure caption for 5a is missing

The authors agree with the suggestions/corrections and these have been incorporated into the text of the paper.

All figure captions were reviewed, and major corrections were applied where necessary.

5a presents the mineral composition of ultrafine iron ore tailings- the redraw the pattern using suitable software and present

Unfortunately, the authors do not have access to the raw data, as the laboratory where the XRD test was performed provided only the image, making the requested correction unfeasible.

Figure 7. Stress-strain curves for different dosages at 14 curing days – no need of markers on the curves

The authors agree with the suggestions/corrections and these have been incorporated into the text of the paper.

Figure 7 has been revised and the markers have been removed, as requested.

The reason for the improvement in strength properties should be explained in detail with literature support

The authors agree with the suggestions/corrections and these have been incorporated into the text of the paper.

Literature support was incorporated throughout the Results and Discussion section to strengthen the findings and substantiate the analyses. In addition, the authors included a statistical assessment based on analysis of variance (ANOVA) and Tukey’s test to validate the results presented.

Reviewer 2 Report

Comments and Suggestions for Authors

The topic is of considerable significance and the authors have prepared a well organized paper.

Please find attached a marked pdf with comments, corrections and a few questions.

Some of the key areas that need particular attention include:

1) Developing the literature review further to establish the outcomes of other studies that have explored polymer based stabilization. On the same topic, more justification should be added to clarify the selection of the acrylic-styrene polymer.

2) Clarification of the usage of the composite i.e. validation of the goals of the study.

2) 

Author Response

Comment

Response

Developing the literature review further to establish the outcomes of other studies that have explored polymer based stabilization. On the same topic, more justification should be added to clarify the selection of the acrylic-styrene polymer.

The authors agree with the suggestions/corrections and these have been incorporated into the text of the paper.

The literature review has been updated to incorporate findings from previous studies on polymer-based stabilization. Furthermore, the Introduction has been strengthened to emphasize the innovations of this work, including the challenge of using polymers to stabilize ultrafine tailings, the development of a predictive strength model based on the porosity/volumetric binder index (η/Pᵢᵥ) concept—which until now has been applied primarily to cement-stabilized soils—and the demonstration of microstructural stabilization mechanisms between the polymer and tailings using SEM and μCT. Such microstructural investigations are scarce in the literature, as most studies focus solely on macroscopic aspects.

Clarification of the usage of the composite i.e. validation of the goals of the study.

The authors agree with the suggestions/corrections and these have been incorporated into the text of the paper.

The authors have added, in the Abstract, Introduction, Results, and Conclusion, indications of potential applications of the composite developed in this research, such as backfill, road base, or sub-base layers. It is emphasized, however, that the main objective of the paper is to elucidate the stabilization mechanisms of tailings with a commercial polymer, from both hydro-mechanical and microstructural perspectives.

Reviewer 3 Report

Comments and Suggestions for Authors

In response to the problem of poor mechanical properties, large production, and adverse effects on the environment and resources of ultrafine iron ore tailings, this study conducted research on improving the physical and mechanical properties of ultrafine iron ore tailings based on polymer stabilization methods. With a focus on developing their potential reuse in civil construction. The experimental design and conducted work are generally comprehensive. However, several revisions are recommended to improve clarity, depth, and academic rigor. Below are specific comments and suggestions for revision:

  1. Suggest the authors to modify the title of the paper, as the current title appears to be a review paper title rather than a research paper title.
  2. In the abstract section, the conclusions need to be more specific, for example, -with 30% and 40% identified as optimal for strength enhancement, what is the specific optimal strength? -Unconfined compressive strength (UCS) tests showed significant improvements in both strength and ductility, what is the degree of change in strength and ductility, and so on.
  3. In the Introduction section, please strengthen the discussion of existing stabilization techniques (e.g., cement, lime) and clearly articulate how polymer stabilization offers advantages (e.g., ductility, lower environmental impact) compared to traditional methods. This will better justify the need for polymeric stabilization. At the same time, the statement in Introduction section is not focused, and there is a little summary of polymer stabilization technology. The text corresponding to equations 1-3 is about cement stabilized soil, which is not related to the research topic. The introduction lacks coherence between the text, and the logic needs to be strengthened.
  4. What are the economic and environmental impacts of polymer stabilization technology compared to traditional methods? Lack of quantitative analysis.
  5. There should be some textual explanation for Figure 1.
  6. Scale bar should be added in Figure 2 (a).
  7. Figure 5 (a) has low resolution and cannot be seen clearly. It should be redrawn using the original test data.
  8. Error bars need to be added to some of the figure data to demonstrate the stability and reliability of the test results, as shown in Figure 9. For other test results represented by lines, the author needs to consider how to represent the error line of the results.
  9. Figure 3 is incomplete. According to the title, there should be three types of images: a) sample; b) two-dimensional image; c) section area.
  10. -From a moisture content value of 15% and an initial dry density of 1.91 g/cm³, How were these two dosages obtained?
  11. The marking line in Figure 12 (d) is very arbitrary.
  12. Please reduce the number of conclusion points and enrich the conclusion text for each point.

Author Response

Comment

Response

Suggest the authors to modify the title of the paper, as the current title appears to be a review paper title rather than a research paper title.

The authors agree with the suggestions/corrections and these have been incorporated into the text of the paper.

The suggestion was accepted and the paper title was changed from “Challenges in the stabilization of ultrafine iron ore tailings using polymeric solution” to “Stabilization of Ultrafine Iron Tailings with Acrylic-Styrene Copolymer for Sustainable Geotechnical Applications”, which the authors believe synthesizes the proposed work more effectively.

In the abstract section, the conclusions need to be more specific, for example, -with 30% and 40% identified as optimal for strength enhancement, what is the specific optimal strength? -Unconfined compressive strength (UCS) tests showed significant improvements in both strength and ductility, what is the degree of change in strength and ductility, and so on.

The authors agree with the suggestions/corrections and these have been incorporated into the text of the paper.

We thank the reviewer for the suggestion to improve conciseness and present the results more specifically, including the UCS values from 49 kPa from untreated material to 2,114 kPa and 3,324 kPa for 30% and 40% polymer content, respectively, the correlation coefficient (R² ≥ 0.90), and the permeability coefficient on the order of 10⁻⁶ cm/s. Furthermore, the abstract has been significantly shortened to under 200 words, in accordance with the journal’s guidelines.

In the Introduction section, please strengthen the discussion of existing stabilization techniques (e.g., cement, lime) and clearly articulate how polymer stabilization offers advantages (e.g., ductility, lower environmental impact) compared to traditional methods. This will better justify the need for polymeric stabilization. At the same time, the statement in Introduction section is not focused, and there is a little summary of polymer stabilization technology. The text corresponding to equations 1-3 is about cement stabilized soil, which is not related to the research topic. The introduction lacks coherence between the text, and the logic needs to be strengthened.

The authors agree with the suggestions/corrections and these have been incorporated into the text of the paper.

The literature review has been updated to incorporate findings from previous studies on polymer-based stabilization. The justification regarding Equations 1–3 focuses on cement because the methodology was originally developed for cemented soils and due to the scarcity of studies applying it to mining tailings, which has been emphasized in the revised text.

Furthermore, the Introduction has been strengthened to emphasize the innovations of this work, including the challenge of using polymers to stabilize ultrafine tailings, the development of a predictive strength model based on the porosity/volumetric binder index (η/Pᵢᵥ) concept—which until now has been applied primarily to cement-stabilized soils—and the demonstration of microstructural stabilization mechanisms between the polymer and tailings using SEM and μCT. Such microstructural investigations are scarce in the literature, as most studies focus solely on macroscopic aspects.

What are the economic and environmental impacts of polymer stabilization technology compared to traditional methods? Lack of quantitative analysis.

Since the focus of the paper is not on economic aspects, any attempt to provide quantitative estimates would be inaccurate and beyond the scope of the study. Nevertheless, the advantages of using polymer stabilization compared to traditional agents, such as cement, include ease of transport—since the amount of polymer is calculated relative to water and can be diluted on-site, thereby reducing transportation costs—its organic nature, which minimizes potential contamination, and, compared to cement, a manufacturing process that is less harmful to the environment.

There should be some textual explanation for Figure 1.

The authors agree with the suggestions/corrections and these have been incorporated into the text of the paper.

The contextualization of Figure 1 has been incorporated into the text.

Scale bar should be added in Figure 2 (a).

In fact, the pen placed next to the tailings was used solely to provide a visual reference for particle size. Nevertheless, the particle size distribution curve provides the most accurate representation of the tailings particle sizes.

Figure 5 (a) has low resolution and cannot be seen clearly. It should be redrawn using the original test data.

Unfortunately, the authors do not have access to the raw data, as the laboratory where the XRD test was performed provided only the image, making the requested correction unfeasible.

Error bars need to be added to some of the figure data to demonstrate the stability and reliability of the test results, as shown in Figure 9. For other test results represented by lines, the author needs to consider how to represent the error line of the results.

The authors agree with the suggestions/corrections and these have been incorporated into the text of the paper.

To comply with the reviewer’s request and enable a comprehensive analysis of the data, the authors performed a statistical analysis using analysis of variance (ANOVA) on the UCS values. The ANOVA results indicated that at least one of the treatments differed at a significance level of 0.05. Accordingly, Tukey’s post-hoc test was conducted to identify which specific groups exhibited significant differences. These results are summarized in Table 3, along with the standard error of the UCS values.

The statistical analysis shows that increasing the dry unit weight to 2.01 g/cm³ for the 30% polymer composite (T-P30/2.01) yields an unconfined compressive strength comparable to that of the 40% polymer composite at 1.91 g/cm³. This equivalence suggests that, from a practical perspective, an evaluation of which parameter—polymer content or dry unit weight—minimizes project cost is warranted. Moreover, increasing the dry unit weight from 1.96 to 2.01 g/cm³ did not produce statistically significant differences, indicating a mechanical response threshold at 30% polymer dosage. In contrast, for the 40% polymer composites, the mean UCS values differed significantly across the three dry unit weight treatments, indicating well-defined performance tiers under the tested conditions.

Overall, the analysis is consistent with the predictive model for the strength of polymer-stabilized tailings based on the porosity/volumetric binder index (η/Pᵢᵥ), which strengthens the study and corroborates the analyses performed from multiple perspectives.

Figure 3 is incomplete. According to the title, there should be three types of images: a) sample; b) two-dimensional image; c) section area.

The authors agree with the suggestions/corrections and these have been incorporated into the text of the paper.

All figure captions were reviewed, and major corrections were applied where necessary.

From a moisture content value of 15% and an initial dry density of 1.91 g/cm³, How were these two dosages obtained?

Initially, an attempt was made to obtain the compaction curve of the tailings. However, it was observed that the addition of water produced a slurry that impaired and hindered the workability of the pure tailings. Consequently, the moistened material seeped through the gaps between the spacer disc and the mold, preventing the application of the standard procedure prescribed by the relevant standard. This finding underscores the challenge of applying the conventional soil mechanics methodological framework to tailings mechanics. Therefore, a moisture content of 15% and an initial dry density of 1.91 g/cm³ were set based on a trial-and-error approach, allowing the tailings to be structured upon water addition, while ensuring that the material did not reach a slurry-like consistency.

Regarding the experimental conditions for the dosages of polymer-stabilized tailings, they were established based on previous studies involving polymer-stabilized soils (Barreto et al., 2022; Boaventura et al., 2023), as well as prior research on the stabilization of iron ore and gold tailings with polymers (Alelvan et al., 2022; Carneiro et al., 2025). Drawing on this theoretical foundation and preliminary tests, a working range of 10–60% polymer, relative to the amount of water to be added, was defined.

The marking line in Figure 12 (d) is very arbitrary.

The authors agree with the suggestions/corrections and these have been incorporated into the text of the paper.

The line was distorted during formatting and has been adjusted. The marking line clearly shows a cluster of tailings particles bound together by the polymer.

Please reduce the number of conclusion points and enrich the conclusion text for each point.

The authors agree with the suggestions/corrections and these have been incorporated into the text of the paper.

We sincerely thank the reviewer for this valuable suggestion. We agree that a more concise and impactful conclusion would significantly strengthen the manuscript. In response, we have completely rewritten the Conclusion section.

The original eight points have been consolidated into four key, overarching conclusions. Each point has been substantially expanded to not only state a finding but also discuss its significance, underlying mechanisms, and broader implications for the field. The revised conclusions now provide a more coherent and compelling summary of the study's contributions, highlighting:

The optimal polymer dosage and the transformative improvement in mechanical performance.

The development and utility of a predictive model for strength based on the porosity/volumetric polymer index.

The microstructural mechanisms underlying the stabilization process.

The environmental compatibility and practical feasibility of the proposed stabilization method.

We believe that the revised conclusion is more focused, insightful, and better reflects the importance of our findings.

Round 2

Reviewer 1 Report

Comments and Suggestions for Authors

The revised version of the paper "Challenges in the stabilization of ultrafine iron ore tailings using polymeric solution" is recommended for publication. 

Author Response

Dear reviewer,

Thank you for your kind approval.

Reviewer 3 Report

Comments and Suggestions for Authors

The authors have made relatively detailed revisions to the manuscript, and the overall presentation quality of the paper has been significantly improved. One suggestion is proposed: in future paper revisions, the authors are advised to highlight the revised text in the revised draft. Otherwise, it will be difficult for reviewers to locate the corresponding revised sections. In addition, the authors’ interpretation of the XRD pattern is unreasonable. The authors claim that the testing institution only provided the image result, however, image results are also derived from raw data, and the storage device of the testing equipment must contain the original data. This XRD pattern remains completely unchanged, which is inappropriate—since the text on the pattern is completely unreadable, and this also does not conform to the general presentation standards for XRD results. Furthermore, the title of Figure 4(a) is incorrectly labeled as "DRX" (instead of "XRD").

Author Response

Dear reviewer,

Thank you for your advice. We were able to find the raw data and improved the image.